# Control of Platform Monopolization in the Digital Economy: The Implication of Open Innovation

**Sergey Silvestrov \*, Vladimir Starovoitov, Dmitrii Firsov and Yuri Krupnov**

Institute of Economic Policy and Economic Security Issues, Financial University under the Government of the Russian Federation, 125167 Moscow, Russia; starovoytow@bk.ru (V.S.); dmitrii-firsov@list.ru (D.F.); krupnov-y@bk.ru (Y.K.)

\* Correspondence: silvestrov-sergey@mail.ru

**Abstract:** This study aims to develop scientifically sound proposals and recommendations for implementing the principles and characteristics of digital monopolies and determine the patterns and relationships of individual products. To achieve the goal, methodological approaches used included correlation analysis. The results are methodological justifications to determine the key principles of digital monopolies within the service approach. The scientific novelty of the results is to identify the synergetic relationship of individual products of digital companies, which can be used to build an appropriate antimonopoly policy of the United States. The practical significance of the results consists of approbation of the proposed principles of digital monopolies on the example of the relationship of Google's selected products.

**Keywords:** digital monopolies; IT corporations; Alphabet; Google; YouTube; open innovation

## 1. Introduction

The relevance of this study is due to the growing crisis of the global economy and the processes of convergence of digital products of technology companies, which force a new look at the processes of antimonopoly policy formation (both at the level of individual states and the level of international organizations). The international practice of antitrust laws proceeds from the prerequisites of industrial society, in which monopolies operate and compete at the level of real goods. Today's realities require a fundamentally new understanding of the relationship between the products and services of technology companies. The end of the twentieth century coincided with global changes in the global division of labor and a new technological revolution in telecommunications and information technology. The Western model of mass consumption society has created the so-called "service economy," where the information services sector has become the driving force of development. Network technology has created a new global marketplace that has directly changed the principles of international trade. Thanks to globalization, which exploits the combined competitive advantages of different countries, new technological chains have emerged that are controlled not by individual national economies but by transnational corporations that have concentrated all the means of production in their hands, including capital and knowledge.

In such conditions, it is no longer possible to speak exclusively about state or national interests since entire countries or regions are often dependent on large supranational economic structures' actions and plans, which can significantly influence both society and international politics. Within such a global stratification of the international economy, the key dominants of economic development are technological companies, whose main product is software. However, despite the ongoing transformation and the increasing role of their global economic impact, to date, there is little understanding of how to build an antimonopoly policy in the current conditions.

The study is relevant because there are still no specific methodologies for determining monopoly predominance. The existing methods and practices originate from archaic precedents and are based on the primacy of the product concerning the information. As was indicated above, modern digital transnational corporations control information flows and distribution channels, thus often starting to infringe on the opportunities and interests of companies and countries that have no direct contact or mutual influence with these information TNCs.

From a scientific point of view, a multinational corporation has direct control over production processes, services, industrial sales markets, and finances compared to an ordinary company in a market environment. In this regard, the issue of antitrust regulation requires innovative approaches not only to how monopolies are regulated but also to how monopolies are defined. Along with the technology corporations and their market prevalence came entirely new and unprecedented methods of quantifying financial markets and commodity markets. It can be assumed that one factor in the success of digital corporations was the fact that there was a convergence of analog products, financial markets, and the digital environment, which created an upward flow of opportunities to capture market power. The practical value of the study is that it determines the specific indicators and parameters of the mutual influence of Google products, through factor models. The factor dependencies may suggest we synthesize new scientific concepts based on market information of existing antitrust laws and quantitative research methods.

If the problems of digital monopolization are discussed, it is important to note such emerging effects of monopolies as a reduction in market competition in the aspect of dumping by large companies of prices for goods and delivery of these goods and services. This effect harms local producers and businesses. Moreover, having a larger market share, digital giants have more metadata on supply and demand, which allows much more flexibility to regulate their supply of goods and services for more demand.

The research question posed by the authors of the study is to determine the limits of the application of factor models in the analysis of large technological transnational companies and their market position. Applying quantitative methods of analysis, we can use the existing market metadata in the context of antitrust legislation.

This study aims to propose methodological principles based on which it is possible to assert the presence of monopoly definitively. At the same time, the authors test the proposed principles of digital monopolies on the example of the relationship of individual products of Google.

The authors address the following theoretical and applied problems as part of their study:

- To formulate what digital monopolies are and their fundamental differences from other types of monopolies.
- To define the main limitations and problems in the sphere of antitrust laws for technological companies.
- To propose the principal provisions characterizing the digital monopolies.
- To determine the actual position of Google in the market of digital products and services based on the proposed characteristics.

This study's hypothesis is that antitrust laws' existing methods and practices have significant limitations to new, innovative products and technology companies, and new principles of attributing a company to digital monopolies will improve the effectiveness of antitrust laws.

If the connections between individual digital products and market patterns are to be assumed, then these connections must be traceable with the help of a mathematical method of analysis. In this case, utilizing the Varimax factor analysis and the principal component method.

## 2. Literature Review

Modern technology companies are characterized by a universal and unprecedentedly high level of involvement in all economic processes of modern society, regardless of country

or industry [1]. However, despite their dominant position, they have successfully managed to avoid the limitations typical of monopoly companies. This is partly because their rapid growth and specificity of product organization for a long time positioned them as innovative and self-organizing systems, whose legislative regulation could negatively affect not only the development of advanced technologies [2] but also the competitiveness of the entire American economy [3,4].

One of the most pressing issues of the impact of these companies on society and the economy is whether these companies can be considered monopolies and whether they fall under current antitrust laws [2,5–7]. This issue is interesting not only from an academic point of view but also an important element of a much broader practical format within the digital phenomenon of the 21st century [5].

Considering the U.S. legal precedents, let us note that antitrust laws do not consider the mere possession of monopoly power in any market to be illegal if it is the result of the legal actions of a particular company. U.S. courts usually begin by looking at a firm's market share to determine whether a company has a monopoly position in a market. However, U.S. courts have not yet identified thresholds above which a company can be classified as a monopolist. Lawmakers note that although a high share of a company within the relevant market does not always mean monopoly power, the market share is one of the most important factors when considering how likely a firm is to obtain monopoly power [8].

Various researchers analyze the impact of macroeconomic stability and transparent government and antitrust policies on financial market development, using panel data from 113 countries from 2007 to 2017. Analyzing GDP, trade openness, and market size, the researchers conclude that macroeconomic stability contributes to financial market development in both developing and developed countries. In addition, the researchers believe that antitrust policy has a significant impact on the level of corruption and bureaucracy in the country [9].

Other authors have devoted their works to the effects of a vertical merger in the United States. They note that the 2020 Vertical Merger Guidelines, which outline the enforcement policies of the Department of Justice and the Federal Trade Commission regarding vertical mergers, were adopted. The authors conclude that vertical mergers can eliminate the significant transaction costs present in contractual alternatives and note that vertical mergers may be preferable to contractual alternatives [10].

Another study aims to analyze how IT platforms could technically integrate into the structure of mobile ecosystems, transforming the economic dynamics that allow largely closed organizations to compete. The study shows that the shift in the formation of platform IT monopolies is caused by the decentralization of these services, leading to a general technical integration of major digital platforms, such as Facebook and Google, into the source code of almost all applications [11].

German analysts argue several fundamental reasons why IT companies with multiple embedded platforms are becoming monopolies. In many digital platforms (such as social networks or search engines), the benefits of using platforms increase as the number of users increases, creating barriers to entry for competitors with fewer users who do not benefit from these positive network effects to the same extent [12]. Then, there is the lowering of the market of competitors and the emergence of the effect of monopolization and the subsequent growth of barriers for the consumer. The classic scenario is targeted work with the client and determining its image based on requests and personal information, which subsequently serves as a tool for increasing demand and optimizing supply. This phenomenon is called «data-opolies» by the researchers from the Harvard Business Review [13,14]. Thus, the European competition authorities have recently brought actions against four data-opolies: Google, Apple, Facebook, and Amazon (or GAFA for short). The European Commission, for example, fined Google a record EUR 2.42 billion for leveraging its monopoly in search to advance its comparative shopping service.

The Canadian government (Bureau) also pays attention to the growth of the digital monopolization trend and takes measures to demonopolize data.In the Bureau's view, it has

the tools to deal with privacy effects under its non-price effects analysis. In a presentation to the Committee, Anthony Durocher, Deputy Commissioner, Monopolistic Practices Directorate, noted that if companies compete to attract users by offering privacy protection, then this quality could be a relevant factor in assessing anti-competitive activity [15,16].

Otherwise in the USA although monopolies may exist, not every dominant firm will necessarily abuse its dominant position. In the U.S., the protection goes further: monopolies are not liable for being a monopoly, i.e., charging excessive prices, reducing privacy protections, or otherwise degrading quality [17].

In the context of the considered topic, it is impossible not to mention such a phenomenon as "cellophane error," a term common in the antitrust literature. This expression is taken from the famous antitrust case of the 1950s when the Supreme Court of the United States gave an overly broad definition of the relevant market failing to identify the real market power of the largest American DuPont chemical company, which monopolized the cellophane market in the United States [13]. Another study focuses on the "cellophane error" phenomenon noting that markets delineated based on the predominant elasticity of demand are likely to be too small, and the potential for the realization of the company's market power will be overestimated [18].

Several studies are devoted specifically to digital monopolies and their characteristics that reveal the concept of "BigTech" and analyze the financial performance of the seven largest global technology companies: Alphabet (Google), Apple, Amazon, Facebook, Microsoft (USA), and Alibaba, Tencent (China). The authors reflect on the growing influence of these IT giants on the global economy and note that all high-tech IT corporations are engaged in scaling and creating network effects and, accordingly, become stronger as digital platforms grow [19]. The researchers also note that monopoly is a unique feature of capitalism, which in today's realities is an integral part of the current model of "big technology." They concluded in their analysis that Microsoft and Apple have been monetizing their market power longer than their technology competitors, which is reflected in their total financial assets exceeding the assets of the other five IT corporations under consideration by USD 53 billion (as of 2019) [20].

Researchers also note that one of the key characteristics of digital monopolies is that they offer their products for free, which effectively means closing the market to new entrants unless they offer better services (also for free) [21].

The literature review shows that there are currently very few comprehensive studies on digital monopolies as "marketplaces," digital platform monopolies, methods for assessing what can be classified as digital monopolies, and others. Note that within the existing research and literature, we can observe the practice of applying three main metrics to determine whether a company is a digital monopoly or not:

- The company is a major player whose products account for a significant share of users or traffic.
- The company's products have no competitors, or there is little competition on the market.
- The company obtains significant competitive advantages in related products due to its dominant position in one market.

In that regard, this study aims to fill the above gap in the subject area in terms of forming a list of principles, compliance that will make it possible to attribute a company to the category of "digital monopolies".

Primary users and the target audience are represented by the broad international research community as well as federal antitrust agencies and NGOs.

## 3. Materials and Methods

The work is presented in the form of the following structural sections. "Introduction" considers and substantiates the necessity of proper assessment of the company's monopoly position and defines the study hypothesis and objectives. "Literature Review" contains an analysis of the main regulatory and methodological principles of determining the monopoly position of companies in the market and includes a review of the literature on existing

methods of monopoly management (including digital). "Materials and Methods" contain the authors' approach to the definition of digital monopolies, which is further disclosed in the "Results" on the example of data analysis of Alphabet Inc. "Discussion" makes it possible to see whether the results obtained correspond to the study hypothesis, and "Conclusion" provides a brief presentation of the authors' findings obtained in the course of the study.

The methodology of this study can be represented as the following scheme (Figure 1):

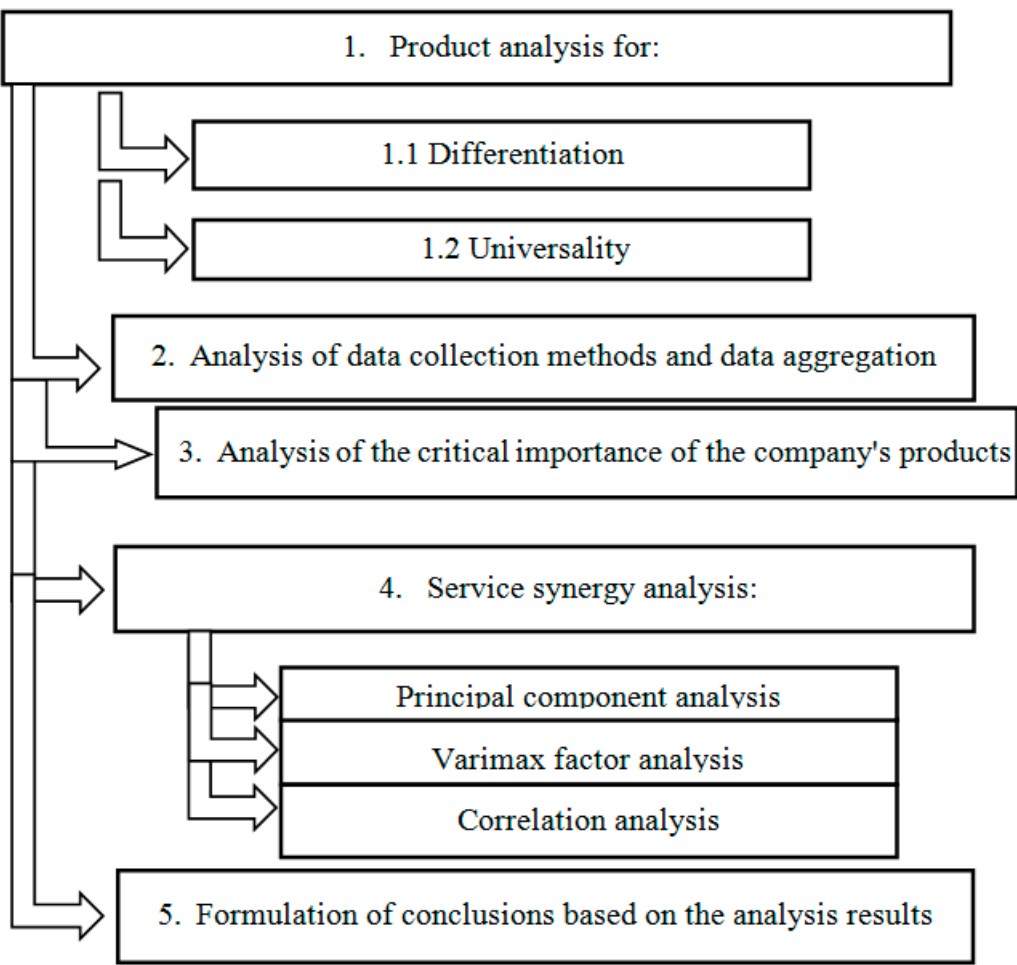

**Figure 1.** The methodological framework of the study.

Considering digital monopolies as a natural element of the digital economy and given the desire of any large technology company to monopolize, a natural question arises as to how and under what conditions a digital company becomes a "platform" digital monopoly. Note that the current evaluation practice lacks studies and materials explaining the type and level of transformation of digital companies into "platform" digital monopolies and the key elements through which they achieve market superiority. The modern system of socio-economic relations is characterized by a high concentration of market power in the hands of digital technology companies. Through their general economic dominant position, they begin to control the key pricing mechanisms of goods and services. Within the existing paradigm of antitrust regulation, there is no understanding of how to expand the methodological apparatus and definition of "monopoly power" from the position of characteristics unique only to large technological companies [22]. Today there is little understanding of how monopolies in digital products and services affect the development and operation of the rest of the economy nationally and internationally. Unfortunately, the practice of modern U.S. antitrust laws continues to be guided by principles rather

typical for the industrial monopolies of the early 19th and 20th centuries, whose position can be determined by simple vertical and horizontal market interrelationships. We can observe a new level of synergy within digital products, whose cumulative effect generates a fundamentally new level of impact on the economy. This synergy forces us to speak about the need to identify digital companies' fundamental elements and characteristics. Based on them, we could speak about the presence of special relationships of these companies' products, which statistical and factor analysis methods can check. In this study, the authors propose four principles that characterize the unique market aspects and characteristics of digital monopolies:

- Principle of differentiation and universality—although company products are differentiated from each other, access to them (or at least to the key majority of services) is possible through a single universal account.
- Principle of information accumulation—information generated by the user is aggregated and processed in direct connection with his/her data within all available services, web services, and devices.
- The principle of critical importance—the company, its services, and web services are critical in the work of a significant number of companies and individuals.
- Principle of service synergy—service and web services have a direct relationship, which can be determined using a factor, comparative, system (and others) analysis.

The proposed principles should make it possible to formalize existing perceptions of which companies are digital monopolies in terms of their product and service positioning.

Within the problem and to test the formulated principles, the applied materials of the study are statistical materials that are freely available [23]. The statistics are based on the aggregate data collected by Statcounter [23]; the sample exceeds 10 billion page views per month. The statistics are updated every day but are subject to review and revision within 45 days of publication. The main tools for modeling and analysis are Data Science libraries of Python programming language: pandas, NumPy, matplotlib, and math. Methods of comparative, correlation and factor analysis are used in the work.

This study is based on publicly available data about the following Google products: the video hosting system YouTube, the Google search engine, the Android operating system, and the Internet browser Chrome. Data on the market capitalization of Google were used as market indicators. At the same time, to determine the exact mathematical relationships, the authors applied the factor models of Varimax rotation and the principal components method.

Varimax rotation is a statistical method used at the level of factor analysis as an attempt to clarify the relationship between factors. Typically, the process involves adjusting the coordinates of the data obtained from the principal component analysis. The adjustment, or rotation, is designed to maximize the variance common to the items. By maximizing the total variance, the results more discretely represent how the data correlate with each principal component [24].

The principal component method is one of the main ways to reduce the dimensionality of the data by losing the least amount of information. Calculating principal components reduces to calculating the singular decomposition of the data matrix or calculating the eigenvectors and eigenvalues of the covariance matrix of the raw data.

The analysis of the market share of the aforementioned products concerning the market capitalization of Google was carried out. This analysis should show the correlation dependence of the market share (Figure 2) of the products of the company on the share value (Figure 3) of the studied company.

Furthermore, the authors analyzed the value of Google stock from 2008 to 2021.

In the case where the attributes are quantified to calculate the correlation coefficient, it is recommended to use the Pearson square method.

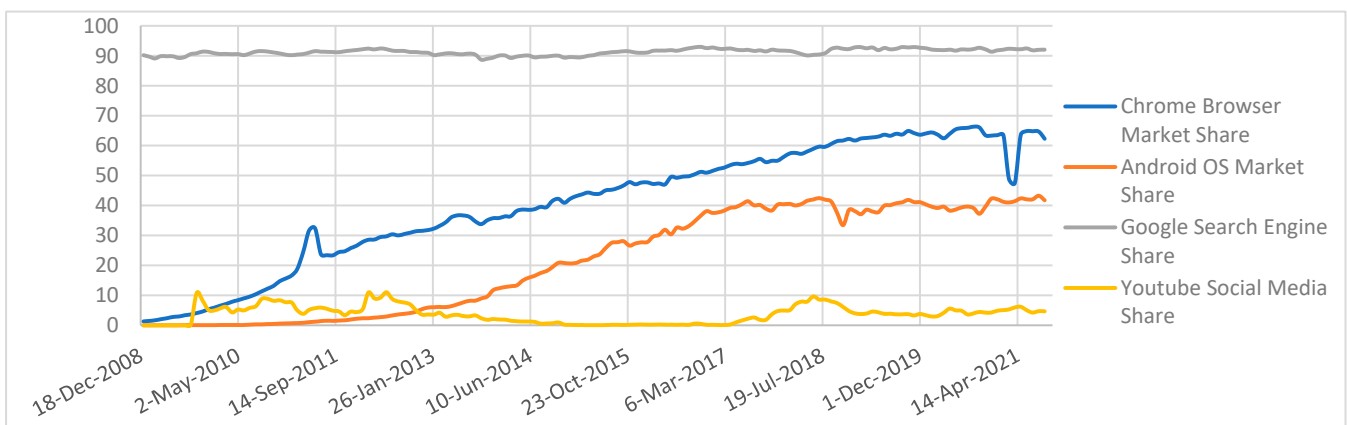

**Figure 2.** Market share of Google companies, %.

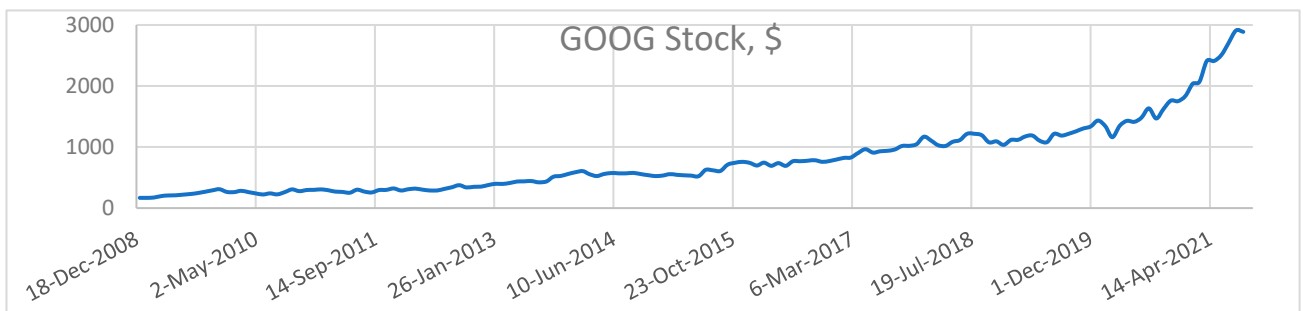

**Figure 3.** Google stock price analysis, USD.

The correlation coefficient in this case is calculated using the Formula (1):

$$r_{xy} = \frac{\sum_{i=1}^{n}\left(d_{xi} \cdot d_{yi}\right)}{\sqrt{\sum_{i=1}^{n} d_{xi}^2 \cdot \sum_{i=1}^{n} d_{yi}^2}} \tag{1}$$

where $d_{xi}$ and $d_{yi}$ are the deviations of each numerical value from the mean value $M_x$ or $M_y$ of its variation series (2), (3):

$$d_{xi} = x_i - M_x \tag{2}$$

$$d_{yi} = y_i - M_y \tag{3}$$

This methodology for calculating the correlation coefficient is embedded in the function corr() of the Pandas library, the multiparadigmatic programming language Python, which the authors used in this paper.

The calculated data are as follows (Table 1):

**Table 1.** Initial data for the following correlation analysis.

|  | GOOG Stock | Android OS Market Share | Chrome Browser Market Share | Youtube Social Media Share | Google Search Engine Share |
|---|---|---|---|---|---|
| count | 152 | 152 | 152 | 152 | 152 |
| mean | 769.6692617 | 21.46440789 | 40.64236842 | 3.642960526 | 91.27835526 |
| std | 551.3412273 | 16.97902121 | 19.64879808 | 2.902345441 | 1.030339392 |
| min | 168.363937 | 0 | 1.31 | 0 | 88.73 |
| 25% | 307.4040145 | 2.365 | 29.2625 | 0.5825 | 90.525 |
| 50% | 582.0295715 | 22.45 | 43.935 | 3.72 | 91.42 |
| 75% | 1087.350006 | 39.185 | 57.7075 | 5.1525 | 92.0825 |
| max | 2909.23999 | 43.26 | 66.35 | 11.04 | 92.99 |

Then, the principal component analysis was applied, which is a method of down weighting. We standardize the original variables so that each of them contributes equally to the analysis. If there are large differences between the ranges of the original variables, those variables with large ranges will dominate the others (for example, a variable that ranges from 0 to 100 will dominate a variable that ranges from 0 to 1), leading to biased results. Converting the data to a comparable scale may prevent this situation. Mathematically, this step involves subtracting the mean value from each value and dividing the resulting difference by the standard deviation. After standardization, all variables will be converted to their original values (Table 2).

**Table 2.** Values obtained by downweighting.

| | | | |
|---|---|---|---|
| 0.50861157 | 0.01690299 | 0.42521816 | 0.72899009 |
| 0.53975102 | −0.18868661 | 0.09431471 | −0.24640463 |
| 0.53428965 | −0.12737301 | 0.11082497 | −0.57354836 |
| 0.00808857 | 0.89642738 | 0.37179379 | −0.22072544 |
| 0.40552713 | 0.37987593 | −0.81226932 | 0.17372614 |

To understand, how the variables differ from the mean for each other, or, in other words, to see if there is a relationship between them, we calculate a covariance matrix (Figure 4) that displays the correlations between all possible pairs of variables in this study.

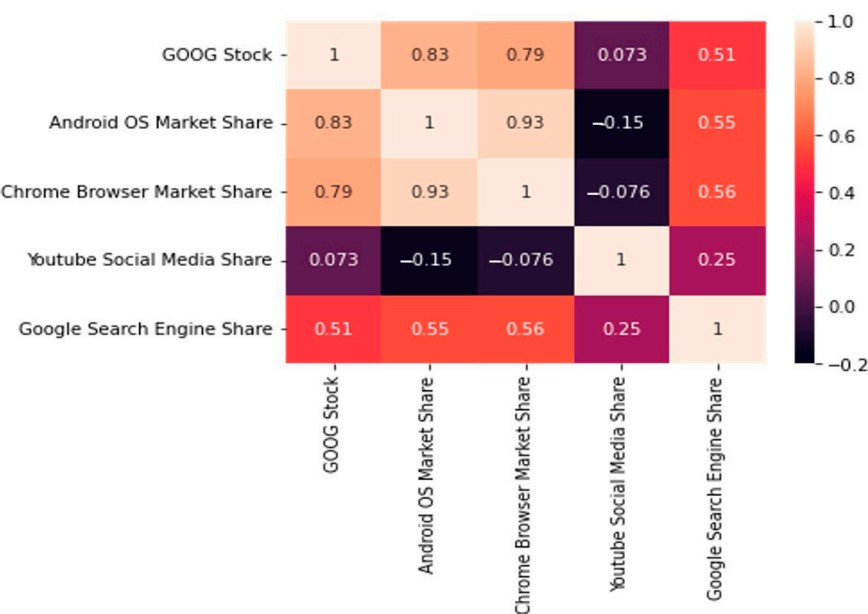

**Figure 4.** Heat map of correlation analysis of Google products [7,10,18,23].

## 4. Results

Let us consider the possibility of applying the presented evaluation principles to the example of Google and its products: the Google search engine, the Android mobile operating system, the social network YouTube, and the Internet browser Google Chrome.

Google is one of the largest technology corporations of the 21st century. The market capitalization of its parent company, Alphabet, exceeds 2 trillion dollars. As one of the very first companies whose core business was formed around the Internet, Google managed one of the first to take a leading position in the search technology market. Subsequent mergers and acquisitions have allowed it to take a significant share of the Internet services market and establish itself as an integral part of the entire web.

- Considering Google from the position of the above principles, we can confidently say that it complies with all of these principles:

- The principle of differentiation and universality—today, Google has 72 independent services and products [25]. Products and services are characterized by a high level of diversification and independence. However, all of them exist in a single accounting ecosystem, within which Google reserves all rights to access and generate data [20].
- The principle of information accumulation—being a cross-platform and multi-service corporation, Google can collect and process data through not only the active use of its services but also through passive methods of analysis [26].
- The principle of critical importance—Google is one of the few companies in the world whose technical failures can have critical consequences for the entire world economy. Moreover, Google independently analyzes and provides public information about its impact on each of the U.S. states [27]. Its impact on the global economy is presented in analytical and research materials of international consulting and rating agencies [1,2].
- The principle of service synergy—the main principle according to which we can say with certainty whether the company has achieved its dominant market position due to individual products or the synergy created through the interaction of many digital products.

In the case of technology companies, the presence of the first three principles makes it possible to screen out services with little influence, concentrating attention on the largest corporations. However, despite this kind of "focusing", there is a chance that corporations do not represent monopolies due to their narrow industry specialization. To determine the synergy of the company's products, let us conduct a correlation analysis of the markets of its main products and stock prices that will allow us to determine the relationships between the products produced, their influence on each other, and the market reaction. The data sample is from 2009 to 2021, and the data source is publicly available on Statcounter.com (accessed on 15 January 2021) (Figure 4).

In the context of the studied company, we can observe a significantly high level of correlation between the products under consideration in terms of the Google Chrome mobile operating system and browser items. The low level of correlation between the Google search engine and the YouTube video hosting site is explained by the very low level of change occurring in these markets. The search giant's share is consistently high, and YouTube's jumping values speak to its mispositioning as a social network.

It is also interesting in what form these products have affected a company's market capitalization over time. Free market mechanisms provide unique tools for assessing a company's competitiveness in its future development (Figure 5).

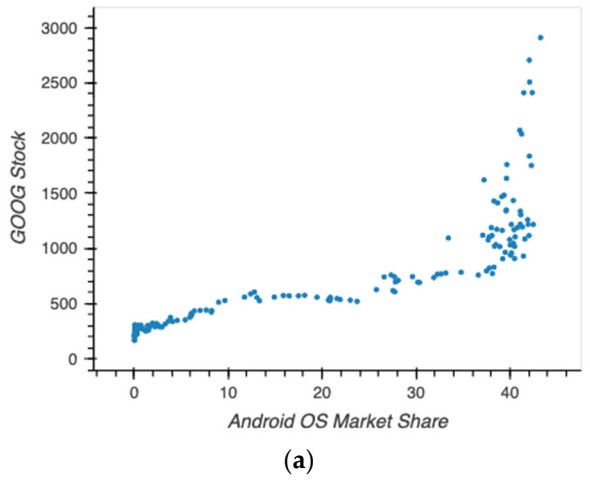

(a)

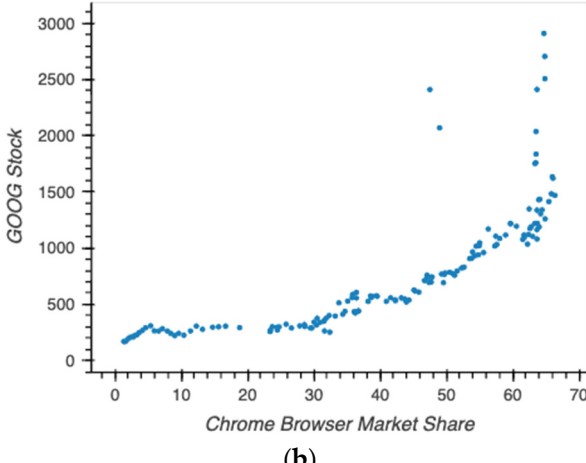

(b)

**Figure 5.** *Cont.*

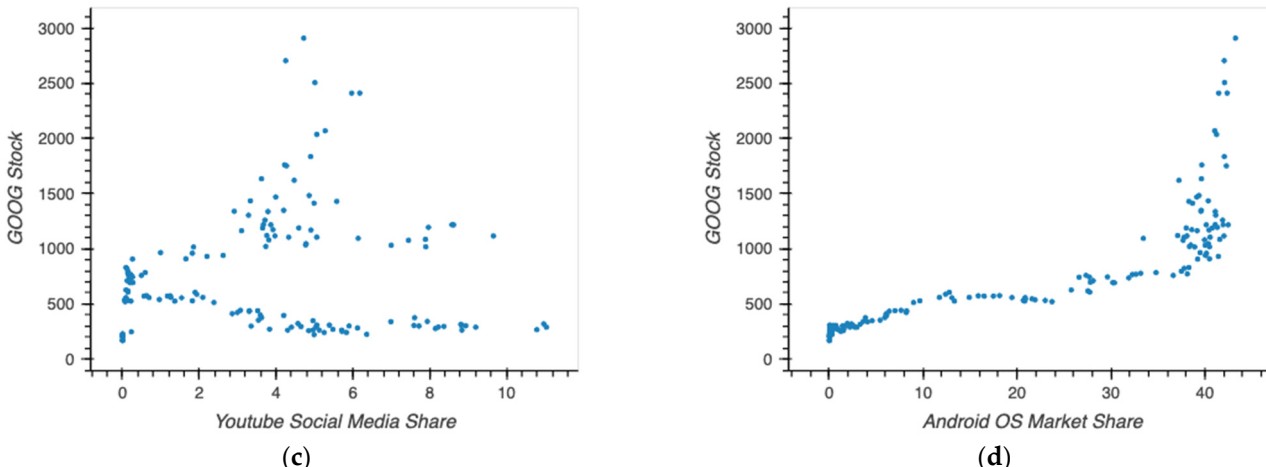

**Figure 5.** Correlation of Google's market capitalization to its products: (**a**) Android OS market share; (**b**) Chrome browser market share; (**c**) YouTube social media share; (**d**) Google search engine share.

Based on the correlations obtained, we can observe a significantly high level of trend fit for all products except YouTube, which corresponds to the parameters of the factor analysis (Figure 6) and is explained by the fact that from the point of view of this assessment it represents an unprofitable business.

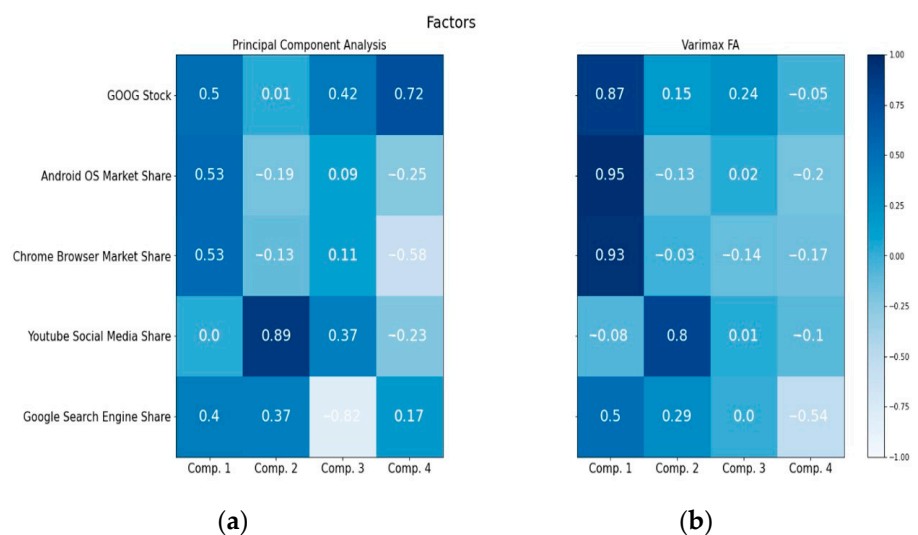

**Figure 6.** Factor analysis: (**a**) Principal component analysis factors; (**b**) Varimax Fa factors.

## 5. Discussion

As noted in the study, current antitrust policy worldwide is characterized by a poor understanding of the internal mechanisms and processes of interaction of different products and services of technology companies. The development of the digital sector of the economy in the last decade has created a new type of corporation, whose influence exceeds in its capacity the previously existing monopolies operated only within certain sectors of the economy and at a much lower level of technological development.

Among the main dangers of digital monopolization, one should mention the ubiquitous collection of data about users for the subsequent optimization of regional offices and a more point-by-point understanding of the business market. It is important to consider this problem from several angles. The first side of this problem is the development of technology sales, the development of consumption and production markets, the optimization of supply chains, and, ultimately, the satisfaction of demand and increased supply. The second side of this problem is the shift in the fundamental elements of stability of

national economic systems. While previously most of such generally accepted indicators of economic development of countries and the same indicators to assess the current level of socio-economic development and forecasting as GDP was based on the commodity income of countries, now increasing attention is paid to the development and implementation of approaches at the national level to the precise definition of the level of well-being by previously indirect, and now direct attributes. Thus, since 2015 Nobel Prizes were given to Angus Deaton in 2014 for his analysis of consumption, poverty, and social welfare, to Richard Thaler in 2017 for his contribution to behavioral economics, to Banerjee, Duflo, and Kremer for their experimental approach to poverty in 2019, and, finally, to Card, Angrist, and Imbens for empirical contributions to labor economics and methodological contributions to cause–effect analysis. To identify and determine these or those regularities in economic development today researchers need data sets, or rather big data sets, and to create approaches to collecting and analyzing these data. Nowadays, it is the dynamics or the rate of change concerning similar periods that are most often investigated; this requires clear and high-quality data sets, which is exactly what transnational digital giants possess and systematically collect. Thus, many search engines, NGOs, business giant manufacturers, and retailers have constantly updated open databases, where every researcher or just interested person can go and get their raw values for the next calculation for free. The main problem is the shadow collection of these data by companies. As a rule, when using those or other mobile applications, users send their data to the company-owner, which then are at risk of being obtained by a third party, not appearing in the contract of offer of the user and the data collector in the person of the company.

The problem with the digital giants and the user data they use lies in their effectively global monopoly position. A user who decides to opt-out of the terms of the user agreement, in most cases, has no alternative. It is not possible to ask a digital platform to remove the clauses of a user agreement, nor is it possible to agree to them in part. We accept them completely or deliberately condemn ourselves to "digital austerity".

It is hard to convince all your friends to leave Facebook and communicate using another social network. It is quite difficult these days to search for information, completely bypassing the services of Google. Nowadays, it is difficult to find a mobile device that functions on an operating system other than Android or iOS. Moving to an alternative platform brings a noticeable inconvenience, and also requires a significant amount of time if it concerns the transfer of data (contacts in the directory, list of friends in a social network, etc.).

On top of that, of course, the digital giants want to maintain and increase their market power. History knows examples of hundreds of so-called predatory takeovers, when a global digital company bought a small promising business or start-up that could potentially be a threat or grow into a fully fledged competitor, with the sole purpose of ending the competing technology's development or taking full control of it to prevent the competitor from growing.

For example, Tobias Blanke and Jennifer Pybus consider digital platforms as a set of services to change the economic dynamics of competition and monopolization in their favor [28]. This paper proves that this shift in the formation of platform monopolies is caused by the decentralization of these services, leading to a general technical integration of major digital platforms, such as Facebook and Google, into the source code of almost all applications.

Now we can safely talk about the rigid predominance of the "platform economy" and as a vector for the development of economic systems and their existing functionality [29,30]. In recent decades, two emergent phenomena have jointly transformed the nature and pursuit of entrepreneurship across industries and sectors: open innovation and platformization. Open innovation involves a shift towards more open and distributed models of innovation, while platformization refers to the increasing importance of digital platforms as a venue for value creation and delivery [31]. Innovation ecosystems are increasingly regarded as important vehicles to create and capture value from complex value propositions [32,33].

Research by Chesbrough views open innovation as a process whereby purposive inflows and outflows of knowledge can accelerate internal innovation thereby creating an opportunity to expand the markets for external use of innovation. One of the main reasons for leveraging open innovation is the fact that one company at a particular point in time may not have all the best brains and skills in a given area of expertise required for innovation. Therefore, partnering with other firms that have similar expertise could contribute to larger outcomes in the long run. This has become an effective business strategy in many sectors today, where open innovation is taken up as a deliberate strategic move [34]

Platform capitalism has created a new dynamic of ever-growing competition and monopolies, as well as technological integration, where industries now depend on each other, despite competing for new customers, lower materials, and costs [35]. When we begin to study this phenomenon closely, we can see that one dominant mode of production depends on different modes of monetization. Thus, we are looking for ways to explore this technical integration and dependency of the platform as a set of services to account for different groups of participants and the infrastructure this supports them within the mobile ecosystem. We see a growing need for a new methodological approach to explore how digital platforms such as Google and Facebook create user services [36].

The market power acquired by digital giants through big data is virtually uncontrollable. There have been sporadic regulatory attempts against global digital companies around the world, and successful cases regarding the misuse of big data to eliminate competition from markets are the exception rather than the rule, even in the most advanced jurisdictions in terms of regulatory development. Germany's competition authority is the furthest along in this regard, with two sets of "digital amendments" to competition law and several cases involving abuse of dominance by digital giants [37,38]. One such case is an antitrust investigation into Facebook for improperly collecting and using user data from other platforms. The decision in that case, however, was later reversed, so this experience, although precedent-setting, can hardly be called a success.

In the authors' opinion, the proposed service principles of digital monopolies will make it possible to determine with greater accuracy the extent to which a particular technology company is a monopolist in the relevant market. The study revealed the current lack of official statistics due to the data closeness because of its commercial importance. The high politicization of technology companies and their influence on public and social processes is also one of the limitations within the topic under consideration [39].

In this context, there is an obvious need for a much deeper understanding of the interrelationships and impact of technology on humanity [40]. Additionally, in the authors' opinion, a fundamental separation of the legislation of technological monopolies from the existing practice of antitrust laws is necessary [41,42]. Development of the international legislation and agreements in the area of limitations can promote the realization of the mentioned provision.

Note that the results obtained allowed us to achieve the goal set and verify the correctness of the hypothesis that within the existing practices of antitrust laws, there are limitations to technology companies, which can be eliminated by implementing the principles developed by the authors.

The practical significance of our findings lies in testing the proposed principles of digital monopolies on the example of the relationship of Google's selected products and confirming their validity through correlation analysis.

## 6. Conclusions

### 6.1. Main Findings of the Present Study

To summarize, we note that this study succeeded in achieving the objectives and obtaining the following practical results:

- The essential characteristics of digital monopolies are formulated, and their differences from other types of monopolies are presented.

- The key limitations and problems in the sphere of antitrust laws for technological companies were defined.
- Principle provisions are proposed that make it possible to characterize the company as a "digital monopoly" in case of compliance with them, namely: Principle of differentiation and universality, principle of information accumulation, principle of critical importance, and principle of service synergy.
- The actual position of Google in the digital products and services market as a technological leader and monopolist is determined based on the formulated principles.

As far as could be determined, IT corporations all aim to establish control over both: their perspective sector of the economy and all down and upstream markets. In practice, it results in significant control of information and associated benefits. Regarding economic and social influence these companies extend power which is not only comparable to the power of the government but sometimes significantly outweighs it due to the specificity of data control.

### 6.2. Comparison with Other Studies

Many researchers describe classical methods of classifying a company as a monopoly and disclose certain characteristics of digital monopolies. However, there are no comprehensive studies devoted to the mechanisms of a clear definition of digital monopolies. The scientific novelty of the results obtained consists of developing integrated principles that make it possible to classify a technological company to digital monopolies with certainty. Additionally, achieved results may establish a foundation for new principles in antitrust laws, based not only on the precedent or the law but on the systemic necessity as well.

The scientific novelty of this work lies in the development of the original tools for analyzing and identifying signs of monopolies in multinational companies developing digital infrastructure.

### 6.3. Implication and Explanation of Findings

The principles proposed in this study can be applied to assess the monopoly position of technology companies in the market of technology products or services. The results obtained on the example of Google show the feasibility of the proposed principles for classifying technology companies as digital monopolies. Sound statistical and factor analysis results indicate the feasibility of necessary changes in antitrust laws. Digital monopolies pose a threat to various intersectoral companies which are influenced by digital data control, as such it is important to identify multiplicative effects and connections between different products of one company, and as far as the results show there is a significant possibility of multiprong influence exerted by even one company.

### 6.4. Strengths and Limitations

It is worth noting that there are no modern methods for assessing a technology company for monopolization of the market "de facto" in world practice. The topic under consideration is new, and in most studies, it is considered only from the position of existing practices. It is also worth mentioning that the high politicization of technology companies creates problems in assessing their actual position in the market of digital products or services.

It is also important to note that any data-driven research requires as much data as possible, especially when it comes to high-tech companies; unfortunately, it is practically impossible to acquire necessary data as they are protected intellectual resource that drives the value-added products of these companies. Peculiarities of Google services and products illustrate how significant their influence is from the market perspective.

The primary limitations of the present study are data driven. Although we use robust factor analysis models, it is very unlikely that they can provide long-term definitive results, it is necessary to broaden the research perspective using new data processing instruments, such as neural networks and AI. Apart from that, it is of the utmost importance

to incorporate additional parameters, such as the number of users and services, number of downloads of the digital products and services, and so on. For now, the lack of these data parameters stands as the biggest limitation of the study.

*6.5. Recommendation and Future Direction*

In the future, it is planned to expand the proposed integrated principles, transforming them into a specific methodology. Based on publicly available data, it will allow for the accurate assessment of technology companies' actual position in the market and, accordingly, identify the degree of their monopolization.

One of the key thrusts for future data-driven research may be found in the utilization of the free market. As for the computational and analytical power, it is unrivaled in its perception of real-world processes, that can weigh in on the future of market relationships and dispositions. Furthermore, it is important to implement a more in-depth systemic analysis of software-related aspects. Modern research lacks nuance in what software development is, mostly focusing on either economic or technical aspects.

*6.6. Limitations and Study Forward*

The study revealed the current lack of official statistics due to the closed nature of the data concerning theircommercial importance. The high politicization of technology companies and their influence on public and social processes is also one of the limitations within the topic under consideration. As far as the research shows, we can definitively outline the importance of future proof methods of research, significant potential of which, can be outlined not only by methodology but also by new data-driven research. Future research should be aimed at solving cross-domain questions and should utilize Big Data and AI instruments.

**Author Contributions:** Conceptualization, S.S. and D.F.; methodology, D.F.; software, V.S.; validation, D.F., V.S. and Y.K.; formal analysis, D.F.; investigation, V.S.; resources, D.F.; data curation, S.S.; writing—original draft preparation, D.F.; writing—review and editing, Y.K.; visualization, Y.K.; supervision, V.S.; project administration, V.S.; funding acquisition, not applicable. All authors have read and agreed to the published version of the manuscript.

**Funding:** This research received no external funding.

**Institutional Review Board Statement:** Not applicable.

**Informed Consent Statement:** Not applicable.

**Data Availability Statement:** Not applicable.

**Conflicts of Interest:** The authors declare no conflict of interest.

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
