# Peer review of "Control of Platform Monopolization in the Digital Economy: The Implication of Open Innovation"

_2199-8531, doi:10.3390/joitmc8020066_

Round 1

Reviewer 1 Report

Dear/s Author/s,

Re: Manuscript “Control of monopolization in the context of the global development of the digital economy: a platform approach to the definition of monopolies”

Reviewer’s report:

The objective of the work is interesting because it tries to propose methodological principles so that the presence of the monopoly in digital companies can be definitively affirmed. The article is well written and structured and shows interesting results related to the synergistic relationship of the products of digital companies for the elaboration of an adequate anti monopoly policy. Likewise, the case study and the methodology are well justified. However, there is no great connection between the Introduction, the Literature Review and the Discussion, nor are the bibliographical references excessive. It is recommended to increase these references so that there is a greater connection between the aspects described at the beginning of this point and the paper can be published.

Best regards

Author Response

Dear reviewers!

Thank you very much for perusal of our work and the recommendations below!

Reviewer report form 1

The objective of the work is interesting because it tries to propose methodological principles so that the presence of the monopoly in digital companies can be definitively affirmed. The article is well written and structured and shows interesting results related to the synergistic relationship of the products of digital companies for the elaboration of an adequate anti monopoly policy. Likewise, the case study and the methodology are well justified. However, there is no great connection between the Introduction, the Literature Review and the Discussion, nor are the bibliographical references excessive. It is recommended to increase these references so that there is a greater connection between the aspects described at the beginning of this point and the paper can be published.

Our answer:

The main literature in subject of «data opolies» is added and different country approaches coping with digital monopolies are marked. The list of references has increased. The following fragment is highlighted with yellow in the manuscript:

 Then there is the lowering of the market of competitors and the emergence of the effect of monopolization and the subsequent growth of barriers for the consumer. The classic scenario is targeted work with the client and determining its image based on requests and personal information, which subsequently serves as a tool for increasing demand and optimizing supply. This phenomenon is called «data-opolies» by the researchers from Harvard Business Review [13] [14]. Thus the European competition authorities have recently brought actions against four data-opolies: Google, Apple, Facebook, and Amazon (or GAFA for short). The European Commission, for example, fined Google a record €2.42 billion for leveraging its monopoly in search to advance its comparative shopping service.

Canadian government (Bureau) also pays attention at the growth of digital monopolization trend and takes measures to demonopolize data. In the Bureau’s view, it has the tools to deal with privacy effects under its non-price effects analysis. In a presentation to the Committee, Anthony Durocher, Deputy Commissioner, Monopolistic Practices Directorate, noted that if companies compete to attract users by offering privacy protection, then this quality could be a relevant factor in assessing anti-competitive activity. [15],[16]

Otherwise in the USA although monopolies may exist, not every dominant firm willnecessarily abuse its dominant position. In the U.S., the protection goesfurther: monopolies are not liable for being a monopoly, i.e., chargingexcessive prices, reducing privacy protections, or otherwise degradingquality. [17]

Reviewer 2 Report

The manuscript discusses the impact of digital monopolies along with their characteristics with respect to the individual products through corelation analysis.

The following points are observed.

  1. The manuscript is well structured and organized.
  2. The motivation and target audience of the study is not very much clear. Who will be impacted with this study?
  3. A detailed background and problem formulation is needed to better understand the work.
  4. Is there any relationship between the GDPR and the digital monopolies?
  5. The strengths and limitation section is very limited. A detailed description can be included. Other parameters like number of users and services, number of downloads of the digital products and services can be explored.

Author Response

Dear reviewers!

Thank you very much for perusal of our work and the recommendations below!

Reviewer report form 2

The manuscript discusses the impact of digital monopolies along with their characteristics with respect to the individual products through corelation analysis. 

The following points are observed. 

  1. The manuscript is well structured and organized.
  2. The motivation and target audience of the study is not very much clear. Who will be impacted with this study?

Our answer:

Primary users and target audience is represented by broad international research community as well as federal antitrust agencies and NGO’s.

  1. A detailed background and problem formulation is needed to better understand the work.

Our answer:

The following fragment is highlighted with yellow in the manuscript:

Then there is the lowering of the market of competitors and the emergence of the effect of monopolization and the subsequent growth of barriers for the consumer. The classic scenario is targeted work with the client and determining its image based on requests and personal information, which subsequently serves as a tool for increasing demand and optimizing supply. This phenomenon is called «data-opolies» by the researchers from Harvard Business Review [13],[14]. Thus the European competition authorities have recently brought actions against four data-opolies: Google, Apple, Facebook, and Amazon (or GAFA for short). The European Commission, for example, fined Google a record €2.42 billion for leveraging its monopoly in search to advance its comparative shopping service.

Canadian government (Bureau) also pays attention at the growth of digital monopolization trend and takes measures to demonopolize data. In the Bureau’s view, it has the tools to deal with privacy effects under its non-price effects analysis. In a presentation to the Committee, Anthony Durocher, Deputy Commissioner, Monopolistic Practices Directorate, noted that if companies compete to attract users by offering privacy protection, then this quality could be a relevant factor in assessing anti-competitive activity. [15],[16]

Otherwise in the USA although monopolies may exist, not every dominant firm willnecessarily abuse its dominant position. In the U.S., the protection goesfurther: monopolies are not liable for being a monopoly, i.e., chargingexcessive prices, reducing privacy protections, or otherwise degradingquality. [17]

  1. Is there any relationship between the GDPR and the digital monopolies?

Our answer:

In case of GDPR regulation of monopolized companies there is often a tension between data protection legislation and business models that depend on personal data as a resource. Uncertainty about what is allowed can have a restraining effect that goes beyond what is necessary to protect the individuals, and it may result in companies applying extra costs to compensate for the increased risk (that will eventually affect the customer).

  1. The strengths and limitation section is very limited. A detailed description can be included. Other parameters like number of users and services, number of downloads of the digital products and services can be explored.

Our answer:

Primary limitations of the present study are data driven. Although we use robust factor analysis models, it is very unlikely that they can provide long term definitive results, it is necessary to broaden research perspective by means of new data processing instruments, like neural networks and AI. Apart from that, it is of utmost importance to incorporate additional parameters, like number of users and services, number of downloads of the digital products and services and so on. For now, lack of these data parameters stands as the biggest limitation of the study.

Round 2

Reviewer 2 Report

The suggested revisions have been incorporated. 

Author Response

Dear reviewer,

Thank you for the overall positive assessment. The spellchecking was performed according to your note.